# A Cloud Information Monitoring and Recommendation Multi-Agent System with Friendly Interfaces for Tourism

**Kune-Yao Chen [1] and Sheng-Yuan Yang [2],***

[1] Department of Information Management, St. John's University, New Taipei City 25135, Taiwan; kychen@mail.sju.edu.tw

[2] Department of Information and Communication Engineering, St. John's University, New Taipei City 25135, Taiwan

* Correspondence: ysy@mail.sju.edu.tw; Tel.: +886-2-2801-3131 (ext. 6396)

**Abstract:** The tourism statistics of Taiwan's government state that the tourism industry is one of the fastest growing economic sources in the world. Therefore, the demand for a tourism information system with a friendly interface is growing. This research implemented the construction of a cloud information service platform based on numerous practical developments in the Dr. What-Info system (i.e., a master multi-agent system on what the information is), which developed universal application interface (UAI) technology based on the Taiwan government's open data with the aim of connecting different application programming interfaces (APIs) according to different data formats and intelligence through local GPS location retrieval, in support of three-stage intelligent decision-making and a three-tier address-based UAI technology comparison. This paper further developed a novel citizen-centric multi-agent information monitoring and recommendation system for the tourism sector. The proposed system was experimentally demonstrated as a successful integration of technology, and stands as an innovative piece of work in the literature. Although there is room for improvement in experience and maybe more travel-related agents, the feasibility of the proposed service architecture has been proven.

**Keywords:** wearable devices; GPS; linked open data; cloud tourism multi-agent systems

## 1. Introduction

Yalçınkaya et al. [1] and the Encyclopedia of the MBA Think Tank (https://wiki.mbalib.com/w/index.php?title=Smart_Tourism&redirect=no) noted that smart tourism is based on a new generation of information technology (also called information and communication technologies, ICT) to meet the personalized needs of tourists and provide high-quality and high-satisfaction services; it has also been considered that smart tourism is a part of a smart earth and smart city, such as that which is described in [1]. Meeting the personalized needs of mass tourists, achieving seamless integration of tourism public services and management, and providing services for enterprises (especially small- and medium-sized enterprises) are the three major goals of intelligent tourism. Most of its core technologies are considered to be the epitome of cloud computing, meaning the Internet of Things (IoT), mobile terminal communication, and artificial intelligence [2], and it is the integration and application innovation of the new generation of information technology. Alongside the tourism statistics of Taiwan's government, the 2017 Tourism Competitiveness Report, as released by the World Economic Forum (https://www.weforum.org/), states that the tourism industry is one of the fastest growing economic sources in the world, accounting for 10% of the global GDP and creating over 300 million jobs.

As Taiwan is facing fast-growing business opportunities created by 10 million potential international tourists, it must consider the future trend of the IoT and develop a complete solution for smart tourism with high added value and a global exemplary effect in combination with Taiwan's advantages in the ICT industry. The handling of unexpected or urgent events involving tourism was the biggest motivation for this study to explore the field of smart tourism in the hopes of autonomously providing corresponding monitoring and recommendation information.

With 2014 as the beginning of the era of the data economy, Gartner (a world-renowned IT industry consultant company that provides an important industry perspective) announced the top 10 technology trends (http://www.gartner.com/technology/research/predicts/), where a clear axis has gradually emerged, including community, action, big data, and the cloud; these four driving forces will be intertwined and change future technological trends. However, according to Ylijoki and Porras (a view of academic trend) [3] and IBM (a view of industry trend), big data has four characteristics (the 4V of big data), the "volume", "velocity", "variety", and "veracity" of data (https://www.ibm.com/analytics/hadoop/big-data-analytics?). Thus, in today's information network era, the characteristics of such data have become more in number, fast, miscellaneous, and difficult to distinguish between real and fake. Therefore, if we can borrow "cloud" technical support; introduce the latest, correct, and complete "open government data"; and provide corresponding information value-added services through "mobile" GPS (global positioning system) equipment positioning technology, the effect of monitoring and recommending the best information as a "community" can be realized. The most important driving force for this research is to cite the four driving forces above and explore a cloud information monitoring and recommendation multi-agent system with both technological trends of industry and academia to meet the real needs of information users.

Open data refers to carefully selected and publicly licensed data, which is not subject to copyright, patent rights, or other regulatory mechanisms, and can be freely published and used by the public without restriction. However, the term "open data" did not appear until recently, due to the rise of the Internet. After the U.S. government established open data government organizations, such as data.gov, relevant open databases have flourished, such as data.gov.uk—the U.K. government's open data website; data.gov.tw—Taiwan government's open data platform, among other databases. Linked open data (LOD) is the vocabulary and technology derived from the linking open data W3C SWEO (World Wide Web Consortium Semantic Web Education and Outreach) Community Project, as initiated by Auer et al. [4], which confirmed that the data are organized according to the linked data principle, and released the information in an open and authorized manner. However, to date, the data formats of Taiwan's government data open platform are mostly XML (Extensible Markup Language), JSON (JavaScript Object Notation), and CSV (Comma-Separated Values, or xls), and the access methods are file download offline or online direct access via the system API (application programming interface), while the latter has a limited number of accesses (about 2000 pieces of data). Therefore, depending on the different data formats and intelligent interfaces for different system APIs, this research proposed a UAI (universal application interface) technology [5] based on the Taiwan government's open data to solve the problem of using pre-storage space and reduce the system processing time. Through local GPS location retrieval, and with the support of OpenData@Taiwan/Taipei and domain ontology, the appropriate open information is compared and intercepted, and then, the corresponding location-based services (LBS) are added, such as providing open information for medical treatment, warnings in case of accidents, or emergencies during travel. This study is based on mobile phone GPS positioning, and the reason why the quality of "integrating linked open data to enhance LBS" should be improved is self-evident (i.e., the recommendation information is credible and correct).

The rise of the IoT in recent years has been accompanied by the popularity of wearable devices. According to Gartner (https://www.gartner.com/en/newsroom/press-releases/2018-11-29-gartner-says-worldwide-wearable-device-sales-to-grow-), a market research organization, 225 million wearable devices will be sold globally in 2019, with an annual growth rate of 25.8% versus 179 million in 2018. In terms of sales amount, wearable electronic devices can generate revenues of USD$42 billion, of

which USD\$16.2 billion is contributed by smart watches. In an information society, where wearable devices are popular, more and more consumers are willing to accept personal health information management with the awakening of their health awareness; therefore, there is a growing demand for the system to provide corresponding monitoring and management information systems. In a report in the Common Wealth Magazine on CES (Consumer Electronics Show) 2016 (https://www.ces.tech/News/Press-Releases/CESPress-Release.aspx?NodeID=2d11a459-e026-409e-a828-3d0e9212cb68), intelligent health networking devices will become the mainstream of products in the future. As the new generation of wearable products become more popular, content will focus more on information services for personal health management; for example, when traveling, the system can provide the location of a nearby sports center. Therefore, this research provides an appropriate personal health information monitoring and consulting agency function in the system through a "wearable device" to further explore how to achieve the related technology of comprehensive intelligent tourism cloud information consultation and a sharing multi-agent system, on the basis of wearable devices, GPS, and linked open data.

On the basis of the aforementioned information and the plans of Taiwan's Ministry of Science and Technology Dr. What-Info (i.e., a master multi-agent system on what the information is): research and development of an intelligent mobile information consultation and sharing multi-agent system (MOST-103-2221-E-129009), and Dr. What-Info II: research and development of an intelligent mobile information consultation and sharing multi-agent system based on Taiwan government open data with universal application interface technology (MOST-104-2221-E-129-009), the relevant completion objectives are summarized and described in [6–8]. This paper further develops a travel cloud information monitoring and recommendation multi-agent system on the basis of wearable devices that include GPS and LOD to explore the LOD access technology suitable for OpenData@Taiwan, thereby integrating this UAI technology chain for wearable devices to increase the accuracy, authenticity, and integrity of intelligent tourism information with the support of three-stage intelligent decision-making, and effectively adding value to the quality of cloud information monitoring and recommendation.

In short, as Eric E. Schmidt, Google's chief executive officer, said at The Mobile First World Asia Pacific event in November 2014, "Action revolution has driven the industry from web first in the past to mobile first. The future is a world of mobile only." This same concept is also mentioned in the article by Shah et al. [9]. This was the initial concept, and it is expected to develop from the relevant practical technologies of mobile devices with the support of the powerful information agent system CEOntoIAS (Cloud Extension of OntoIAS—Ontological Information Agent Shell) [10,11] at the back end for solving the problem that mobile devices cannot provide effective information services due to the lack of computing power. The proposed system has friendly interfaces and can refer to the aforementioned UAI-based LOD access technology in collaboration with GPS location interception, and with the support of appropriate linked open information and domain ontology indexing for autonomously providing corresponding information, as well as effectively improving the monitoring and recommendation qualities of the system's mobile information.

## 2. Literature Analysis and Discussion and Development Technology

Wearable devices, introduced in 2013, contain miniature electronic devices that can sense, wear, display, calculate, and carry out corresponding information and activities through a network. The basic core function includes an attachment device, physiological sensing, activity sensing, and environmental sensing, among others. At present, the relevant application markets include various professional and special fields, such as information, entertainment, health and fitness, medical treatment and care, and safety and security. There is plentiful relevant literature in Taiwan and abroad on this topic, such as Ke's [12] analysis of the management of health willingness through wearable devices and exploration of the development trend of its application in personal health insurance. On the basis of a smart watch life log application for diabetics, Preuveneers and Joosen [13] explored technologies that use cloud and homomorphic encryption to ensure privacy. Adapa et al. [14] examined the factors associated with the adoption of smart wearable devices, especially Smart Glass (Google Glass) and Smart Watch

(Sony Smart Watch 3). Jiang et al. [15] put forward a user-centric three-factor authentication scheme to ensure the privacy and security of sensitive data for wearable devices assisted by a cloud server. As mentioned above, wearable devices are becoming an indispensable part of future human-oriented information services, especially in the personal health sector. For this reason, this study also refers to the personal recording function (e.g., calorie consumption monitoring) of a wearable device, and provides appropriate information services (e.g., corresponding sports centers or places) for the wearer in a travel itinerary, which completely portrays all-around intelligent travel information service.

Open data itself is not a new concept. Generally speaking, the application of open data is mainly non-textual data material, which is often opened because the data have commercial value, or it can become a valuable product after consolidation. LOD is an applied open database organized by the linked data principle. There are also many open data or LOD-related studies in the literature; for example, Yu [16] referred to the traffic class parking lot of the Taipei City open data platform YouBike (YouBike: Smile Bike is a public bicycle rental system in Taiwan), and MRT (Mass Rapid Transit) to explore the related technologies of linked open data application. Alobaidi et al. [17] proposed using LOD to enrich query content and improve the effectiveness and ranking of searches. Musto et al. [18] proposed a methodology to automatically feed a graph-based recommender system with features gathered from the LOD cloud, and analyzed the impact of several widespread feature selection techniques in such recommended settings. Sansonetti et al. [19] described a hybrid recommender system in the artistic and cultural heritage area, which considers the activities on social media performed by the target user and their friends, and takes advantage of linked open data sources. Most of the above studies mentioned value-added follow-up information services with LOD support; under the support of the WIAS (web service-based information agent system) cloud platform [11], this study also referred to UAI technology in order to retrieve and translate according to the local GPS location and the three-tier address-based comparison technology, and intercept appropriate corresponding LOD information to support the operation of overall information processing, exchange, communication, operation, integration, analysis, and decision-making support, thus, enhancing the correctness, authenticity, and integrity of LBS information.

With the advancements of the technology era, handheld devices can receive calls, send and receive messages, and provide music and ringtone downloads, multimedia messages, video calls, and other functions. In addition, information and related services at specific locations are provided through wireless networks, which is called LBS. However, LBS was originally a rescue service provided by U.S. operators in 1996, and at that time was called E911 [20]. The applications of LBS explored in the literature include Hsu [21], who used the service quality model (SERVQUAL) to explore the relationship between Uber's LBS quality and satisfaction, as well as its related applications. Wang et al. [22] introduced the issues of location awareness and privacy protection, and explored technologies related to LBS when users have different requirements based on different locations. Sun et al. [23] introduced LSA (long-term statistical attack) by using historical data, and offered two methods to preserve users' privacy, including dividing the regions of the map into different PLs (levels of protection requirements) according to the privacy requirements, and analyzing the ability to preserve user's privacy by entropy. Jang and Lee [24] presented the features of LBS that influence the usage intentions of users and the moderating effects of innovativeness for LBS in sustainable mobile-related industries. On the basis of the above LBS documents, it is known that the location-based service, mobile positioning service, and location service can obtain the location information (geographic coordinates) of the mobile terminal user through the mobile operator's radio communication network (e.g., GSM- Global System for Mobile Communications network or CDMA- Code Division Multiple Access network), or an external positioning method (e.g., GPS), and provide users with corresponding value-added information services with the support of the back-end GIS (geographic information system) platform. In this study, under the support of the CEOntoIAS cloud platform, the UAI technology was used to retrieve and translate according to the local GPS location and the three-tier address-based comparison technology to intercept appropriate corresponding LOD information; support the operation of overall information

processing, exchange, communication, operation, integration, analysis, and decision-making support; and enhance the correctness, authenticity, and integrity of LBS information, thus effectively adding value to the quality of the cloud information monitoring and recommendation in the system.

Most of the aforementioned systems proposed an appropriate agent system architecture for users of specific jobs and cannot be adapted to the changing operating environment. Moreover, most of them also used a specific language to develop a specific agent system, which has poor functionality expansion and is difficult to maintain. Web service is the technology proposed to solve the above problems, as it is the foundation for a cloud (mobile) information system to be pushed to the extreme. There are many systems that discuss the cloud (mobile) information architecture in the literature; for example, Sun [25] proposed an internet service composition system based on adaptive trust and a reputation model of dynamic social networks to explore how to ensure and monitor the quality and reliability of service composition under the system heterogeneity and dynamic characteristics of the services. Wei et al. [26] explored relevant technologies for building interoperable and extensible text-mining network services on the basis of the concept of exceeding accuracy. Yoshiura et al. [27] developed and implemented an acceptable, useful, and transparent web-based information system for a regional mental healthcare service network in a medium-income country with a decentralized public health system. Ghavami [28] provided an advanced traveler information system on the basis of open geospatial consortium standards to perform fuzzy bi-criteria shortest-path analysis for a multimodal transportation network. This study was also based on the support of web service technology to construct the cloud web service information agent system WIAS, which is designed and constructed to provide the corresponding web services for the operation of related subsystems. Many flexible system designs are also cited, such as standardized and flexible parameter transfer formats, quickly disassembled and reorganized SQL IC (structured query language integrated circuits), and modular network service function design. All these designs positively present the excellent performance of web service technology. A cloud (mobile) intelligent information processing and decision support multi-agent system was also proposed to discuss the feasibility of supporting specific cloud information services, as well as related implementation technologies and issues, in order to achieve the research scope of integrating web service technology and a multi-agent system architecture.

In this era of information explosion and confusion, the demand for information monitoring and recommendation technology is growing, such as that suggested by the systems proposed in [29,30]. Therefore, on the basis of the many practical advances in the research, development, and implementation of the Dr. What-Info system in the early stage, this paper enhanced a multi-agent system with friendly interfaces for intelligent travel cloud information monitoring and recommendation based on wearable devices, GPS, and linked open data. Regarding the aforementioned travel needs, this paper prioritizes the completion of the three most basic and important tourism-related subagents, including a personal health information monitoring and consulting agent (PHI agent), a restaurant information consultation and sharing agent (RI agent), and a beverage safety consultation agent (BS agent), in order to meet the basic travel needs of users for this food kingdom in Taiwan (for example, the recommendation of gourmet restaurants or the monitoring and recommendation of overeating in the face of food). These subagents explore LOD access technologies for integrating the aforementioned UAI technology chain with OpenData@Taiwan for mobile devices to expand the accuracy, authenticity, and integrity of intelligent tourism information with the support of a three-stage intelligent decision-making strategy. This paper not only extends the application scope of Dr. What-Info, it also attains the ultimate goal of this research—the establishment of an intelligent mobile information monitoring and recommendation multi-agent system with the characteristics of being precise, rapid, tough, universal, and having initiative.

## 3. Proposed System Technology, Architecture, and Operation

### 3.1. Domain Ontology and Related Ontology Services

This study used Protégé (http://protege.stanford.edu/) to construct the basic framework of domain knowledge, and used this tool to develop many ontologies, including PC-DIY (Personal Computer Do It Yourself) ontology, researcher ontology, thesis solicitation ontology, network protocol packet format ontology, energy-saving information ontology, and YOHO (Young Home) information ontology, which are used to verify the validity and correctness of the ontology construction methods and tools. Protégé API (https://protegewiki.stanford.edu/wiki/WebProtegeUsersGuide) was used to construct and enhance the processing technologies for the relevant information systems supported by ontology. In addition, this study constructed a reasoning primitive function that supports the application of knowledge ontology. Finally, the ontology architecture was utilized to index actual information for rapid and accurate access to relevant information (see Figure 1 for the index concept). First, Figure 1a shows how the ontology index is used to index relevant files, then to filter the documents that do not contain this ontological word with the partial full-text index, and, finally, to narrow the search scope of the documents in two stages. This approach can accurately judge useful information, as ontology has the precise meaning of domain words, and can quickly index and filter useful information by combining it with real data to reach the goal of ontology, meaning it supports fast and accurate access to useful information (see Figure 1b for details). Furthermore, the system further modularizes the relevant usage information according to different usage situations, which facilitates the quick access of the system, and thus effectively supports the overall system operation, as shown in Figure 1c.

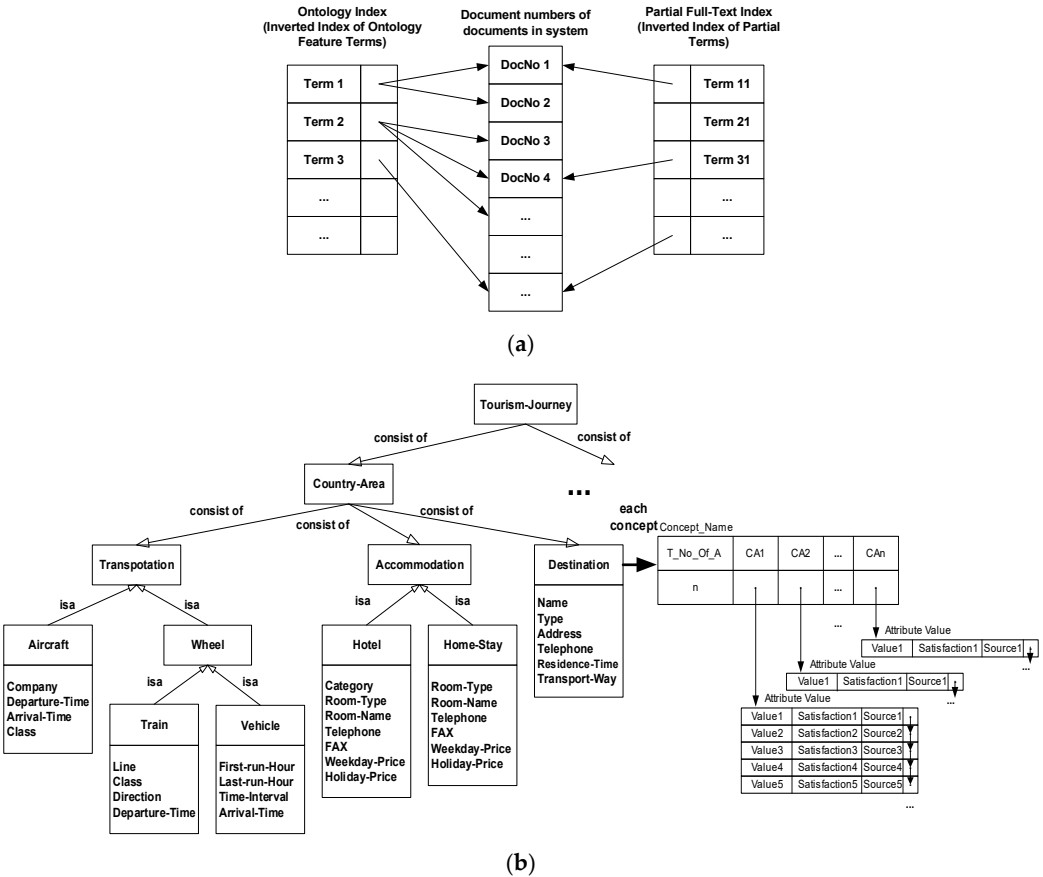

(**a**)

(**b**)

**Figure 1.** *Cont.*

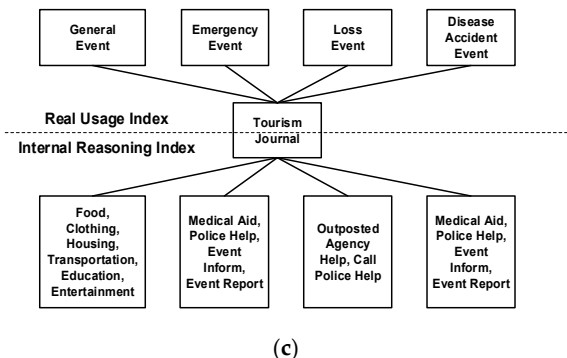

(**c**)

**Figure 1.** Tourism ontology index structure and data relationship: (**a**) a two-stage index structure; (**b**) relationship between ontology index structure and real data; and (**c**) modular connection and use of context index in tourism event ontology.

Ontological services include the semantic distance conversion of search words and conversion services, such as search words corresponding to upper words, lower words, synonyms, and antonyms. This study adopted WordNet (http://wordnet.princeton.edu/) as the base of the comparison model in combination with the Chinese–English bilingual ontology word network (https://ckip.iis.sinica.edu.tw/CKIP/engversion/index.htm) of Taiwan's Academia Sinica to discuss the transformation of Chinese–English bilingual information, the link between language information and the conceptual framework, and the link between word sense differentiation and word sense relationship, as well as their fields of use, in order to provide the basic framework for knowledge management in this system. *Jaccard similarity* (or Jaccard index) was introduced to estimate the consistency between ontological concepts [5,6]. The basic concept is to index the domain concept by using the consistency between the concept of the search term and the corresponding concept of WordNet and its related position. Finally, the domain concept is supported by the identifier Synset_ID in WordNet to support the overall system operation. The most important factor is that users can employ SQL and JWNL (Java WordNet Library (http://jwordnet.sourceforge.net/handbook.html)), to access the WordNet database, which is also the main reason why this study used the SQL database to construct ontology and the Java development system. Furthermore, the position of this domain concept in WordNet ontology is indexed according to *hasURI* of the two attributes of *hasURI* and *hasConsistency* for this most representative domain concept. The consistency of the domain concept is calculated by Jaccard similarity, and stored in the corresponding *hasConsistency* attribute to complete the corresponding processing of the domain concept. Finally, according to the *Synset_ID* of the concept in WordNet, the position of the domain concept is accessed in the ontology database (OD) to support the overall system operation in order to solve the precision, toughness and universality problem of the same meaning in different domains.

### 3.2. Building UAI-Based LOD Access Technology

In the earlier stage of this study, the solution through the Dropbox cloud space, which was researched and developed as an open data platform, and did not provide APIs for accessing the corresponding database, but only provided hyperlinks for downloading the open database. The solution to this issue is UAI technology, and the related steps are described in [5,7]. The establishment of UAI access to the open database by the subsequent system only requires the conversion of the location of the user's local GPS to the corresponding address; it then compares the appropriate open database for the keywords of the user's current event, and then converts the corresponding map display according to the data intercepted in the "address" field to reach the research goal of the aforementioned value-added location-based mobile information service. Furthermore, the back-end system must first build the ontology of the corresponding event (as shown in Figure 1c) and link it to the appropriate open databases. The APIs corresponding to the map are accessed on the basis of the open database query, and the query results are then developed to complete the establishment of the whole UAI-based LOD

databases for achieving the goal of fast processing on the basis of the concept of exchanging space for time.

Regarding the establishment of the UAI-based LOD cloud database, first, the back-end cloud database uses the Parse cloud database to process the JSON (JavaScript object notation) format, and then uses a Google Cloud Spreadsheet to process the CSV (comma-separated values) format. By referring to the aforementioned UAI technology, it processes files or other formats via Dropbox to collate the relevant government open data, and access system-related images and data. In general, the government open database provides two kinds of data link concatenation methods; one is that the API links provided by the government database are directly used, and the other is to download the data for use by oneself. The reason why this study did not choose the former method, which seems more direct and convenient, is that the amount of data used by the system is quite large and its content is more complicated. If the open database is directly connected in series, it will only increase the burden of interpretation and management of the system. Therefore, this system adopted the method of periodically uploading open data to the Dropbox cloud database, which facilitates the management of data categories. However, as Dropbox has not yet provided the relevant API for JavaScript, this system and Dropbox cannot be concatenated directly; therefore, the above UAI technology was used to solve this problem.

This study established a related travel event ontology architecture (as shown in Figure 1c) with a back-end LOD cloud database, allowing the breadth and depth of the system to be changed according to the needs of the system. In other words, the information breadth can be adjusted by increasing or decreasing events, while information depth can be determined by matching it with the provision of back-end data and functions. The correspondence between some events and back-end data is shown in Figure 2. This not only makes it easy for the system to achieve the research and development goal of flexibly tuning the depth and breadth of recommended information, but also enables the event ontology, as established by the system, to provide block information connection according to different event situations, thus improving the system's access efficiency and response time. The system captures the user's position through GPS, combines the corresponding government link open data stored in the back-end cloud database with the relevant event corresponding ontology, and captures and compares the appropriate open information in the administrative area, meaning the location of the corresponding user, in order to provide corresponding location-based information services such as community sharing, parking lot details, emergency dialing, and other value-added functions. Finally, using Google's reverse geocoding function, the corresponding LOD cloud database can be easily accessed by GPS information retrieval to support the subsequent application of the system for effectively and efficiently achieving the concept of exchanging space for time.

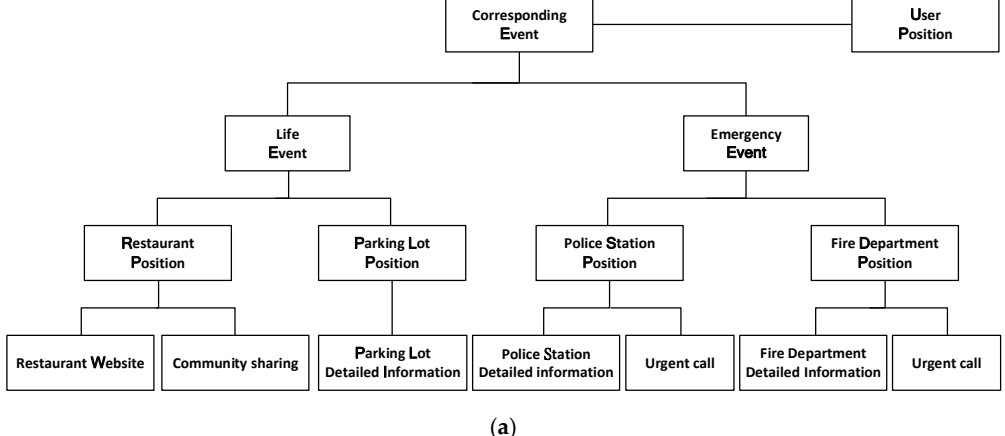

(a)

**Figure 2.** *Cont.*

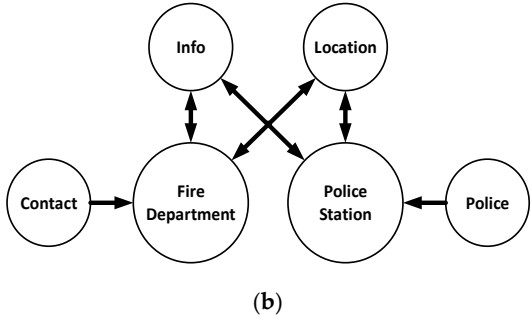

(**b**)

**Figure 2.** Correspondence between some events and back-end data: (**a**) ontology architecture diagram of some tourism-related events; and (**b**) linked open data (LOD)-linked diagram (police and fire service).

### 3.3. Architecture of WIAS

The design concept of this WIAS is a service-oriented architecture (SOA) based on web services, that is, providing relevant information services through the existing network environment and communicating with each other through XML in JSON format. When the required information services are finally assembled and placed on the network for use, it improves the system's performance and development difficulties. The concept of this web service design is to plan related cloud web services on the basis of the services required by CEOntoIAS. The integration includes the web services required by the interface agent (IA), the data mining agent (DMA), and the case-based reasoning agent (CBRA). These factors are responsible for providing cloud data transfer and system calculation for the corresponding agent system, including, offer common cloud information required by the system to access core of this Web service—cloud system database—which is the aforementioned SQL software IC concept that binds different access parameters through SQL access templates and is responsible for communicating with relevant cloud web services and corresponding cloud databases to provide the corresponding cloud information service. With the support of the tourism ontology index structure (see Figure 1 for details), after establishing the cloud information association graph, the corresponding cloud data transfer mechanism must still be designed, including the client or user side (i.e., each agent system), servo side (i.e., WIAS), API information transfer interface, and cloud information core database, which constitute a cloud information transmission mechanism. In other words, each agent program can trigger an event (such as travel) by calling the corresponding cloud system function library (APIs) with different parameters in order to obtain the corresponding information functions for easily building proper information services, as detailed in [11].

Figure 3 shows the WIAS architecture. When the source information is an access command, the corresponding SQL access template is triggered directly through the web service-based interface by using the SQL IC constructor. After binding the relevant access parameters, the corresponding access results are retrieved from the raw database, and then, transmitted back to the interface agent system through the web service-based interface. Regardless of whether the source information is a cloud default solution, it will also be sent through the web service-based interface to query the predefined rule base, and then return the corresponding cloud default solution to the interface agent. However, if there is no default solution, it will trigger OntoIAS (ontological information agent shell) [10] to directly seek the appropriate cloud recommendation information solution from outside the Internet through the technologies of information search, information retrieval, information classification, and information presentation or sorting. In addition, the raw database provides all frequently historical information to OntoCBRA (ontological case-based reasoning agent) [31] as cloud information material for case generation, and infrequently historical information with a two-stage time series exploration algorithm, as the information material that triggers OntoDMA (ontological data mining agent) [32] to produce cloud prediction solutions, where the construction of predefined rules is under the supervision of domain experts. The aforementioned cloud recommendation information solution is determined through rule maker, and the agent system architecture applies information transfer on the Internet

through cloud computing technology for the implementation of an on-line web service interface, which enables agent programs to call a common function library. An overview of some cloud web services' APIs is shown in Table 1. The description specific to UAI includes the four categories of connect, compute, search, and storage [33], which makes it easy to communicate with cloud databases, and the user can easily add and update related cloud information functions, and timely reflect them in the corresponding agent programs, thus achieving the research purpose of a cloud information agent system-based on web service technology.

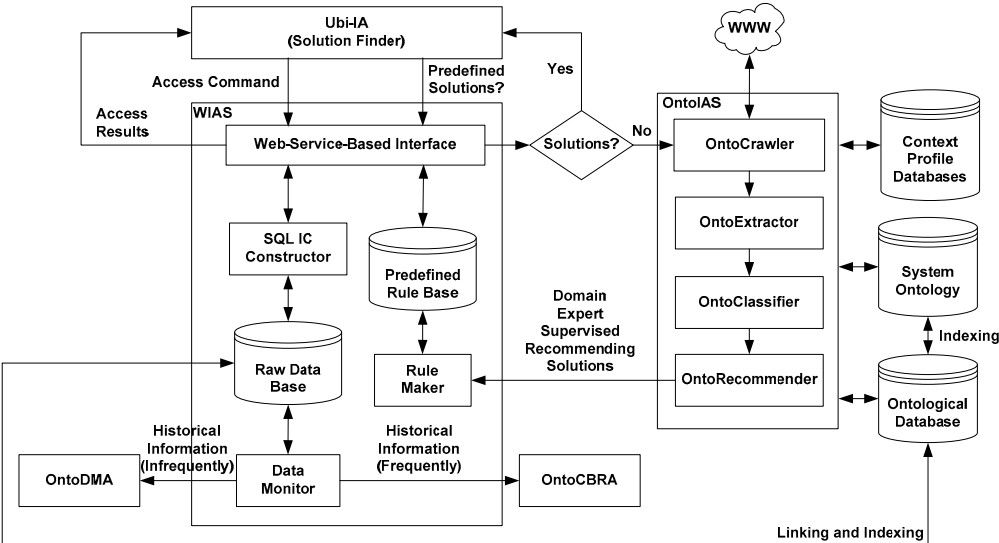

**Figure 3.** WIAS (web service-based information agent system) architecture. Ubi-IA: ubiquitous interface agent, SQL IC: structured query language integrated circuits, OntoDMA: ontological data mining agent, OntoCBRA: ontological Case-Based Reasoning (CBR) agent, and WWW: World Wide Web.

**Table 1.** Overview of some parts of APIs' (application programming interface) functions. Case-Based Reasoning (CBR).

| SERVICE NAME | SERVICE DESCRIPTION | TO WHOM |
|---|---|---|
| CBR_InsTmpCaseData | Adds temporary data for solutions | CBR Agent |
| CBR_Solutions | Solution to case library | CBR Agent |
| DB_InsCaseBase | Adds case base rule | Sharing |
| DB_InsPrediction | Adds a prediction rule | Sharing |
| DB_UptCaseBase | Records whether it been converted | Sharing |
| DB_UptRawData | Records whether it been converted | Sharing |
| DM_Solutions | Solution to prediction rules | Data Mining Agent |
| DM_TransCaseToPred | Converts from a semantic case library to a prediction rule library | Data Mining Agent |
| DM_ViewCaseBase | Views case base | Data Mining Agent |
| IA_InsRawData | Adds original data | Interface Agent |
| IA_Solutions | Solution to redefined rules | Interface Agent |
| Share_DelUniqueTables | Deletes the data in each independent data table | Sharing |
| Share_IsNumeric | Determines whether str NUMBER is a number | Sharing |
| Share_ViewCBR | Views case library summary | Sharing |
| Share_ViewDBSDT | Views system time | Sharing |
| UAI_ConnectCrossDomain | Connects cross-domain Open Data Base | UAI_Connect |
| UAI_ConnectAlertReg | Links registration tips | UAI_Connect |
| UAI_ConnectRuleSetting | Connects rule settings | UAI_Connect |
| UAI_ComputeFileAccess | File access computation | UAI_Compute |
| UAI_ComputeTimeSeriesAccess | Time series computation | UAI_Compute |
| UAI_ComputeTriggerConfig | Triggers configuration computation | UAI_Compute |
| UAI_SearchTimeSeries | Time series search | UAI_Search |
| UAI_SearchBuildIndexFields | Creates an index field | UAI_Search |
| UAI_SearchStatistics | Searches statistics | UAI_Search |
| UAI_StorageFFStatus | Folder and file status | UAI_Storage |
| UAI_StorageFFManipulation | Folder and file operation | UAI_Storage |
| UAI_StorageFTransimission | Folder and file transmission and status | UAI_Storage |
| UAI_StorageFFSharing | Folder and file sharing and integration | UAI_Storage |

### 3.4. Design and Construction of Dr. What-Info System with CEOntoIAS

The tourism-related ontology (Figure 1) technology can help users quickly, accurately, and effectively obtain appropriate and timely cloud information and use the OntoIAS for allowing us to accomplish the research purpose of expanding the information agent shell of EOntoIAS to CEOontoIAS (refer to Figure 4 for details). The support includes: (1) driving the intelligent cloud information data mining agent system OntoDMA [8] to utilize the classification and clustering technology of cloud data mining; extracting the relevant cloud information operation knowledge or rules according to the relevant support and confidence, as provided in advance by the system cloud operation mode; and actively providing the system with suitable, real-time, and fast cloud information prediction service; (2) driving the intelligent cloud information case reasoning agent system OntoCBRA [31] to match the cloud information ontology service and similarity calculation between the corresponding cases, introducing a case-based reasoning mechanism, actively learning the relevant system knowledge, and expanding the cloud system knowledge and operation mode, and thus enhancing system-related cloud information processing robustness; (3) when neither of the aforementioned factors can provide an appropriate cloud information solution, the system triggers OntoIAS [10] through the solution finder to search for the appropriate cloud information solution from outside the Internet directly through information search, information retrieval, information classification, information presentation or sorting, and other technologies. This solution is passed to the front- and back-related cloud information processing systems via Ubi-IA (ubiquitous interface agent) with a canonical user request representation language (CURRL) [10] in JSON format through the solution finder, becoming the basic cloud information material for OntoDMA and OntoCBRA to further expand learning prediction and case-based reasoning, which completes the learning cycle of the whole system in response to the information query and enhances the robustness of system cloud information processing and decision-making support. The CURRL is a frame-based command representation that makes it easy to map query command intentions, objects, and goals into corresponding frame slots. Table 2 shows the frame slots and their descriptions, whereas Table 3 illustrates some examples of query commands. Finally, through the cloud information processing of OntoDMA, OntoCBRA, and OntoIAS three-stage intelligent decision-making, as well as with the support of the cloud intelligent multi-agent system shell CEOntoIAS (which supports the processing, exchange, communication, operation, integration, analysis, and decision–making of cloud information), the research goal of autonomously searching and recommending the best cloud information solution can easily be reached.

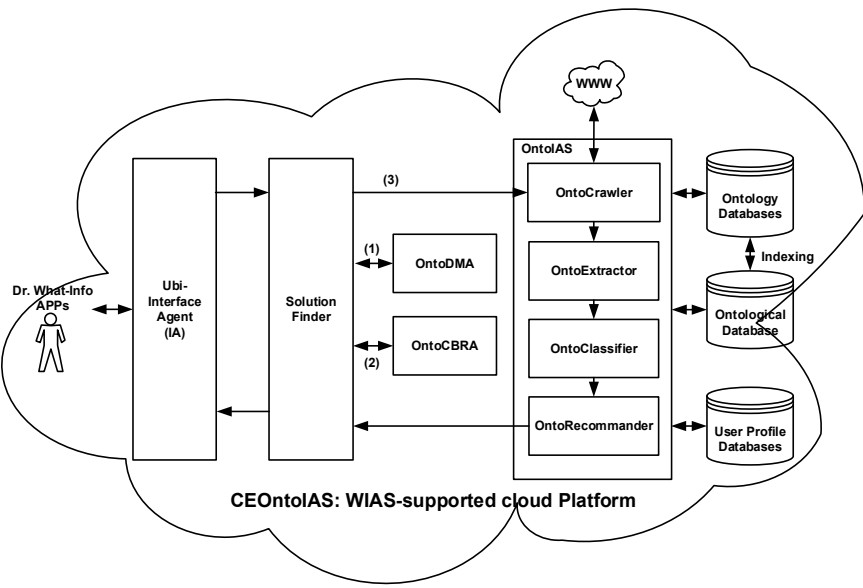

**Figure 4.** Dr. What-Info system architecture.

**Table 2.** Frame structure of canonical user request representation language (CURRL).

| Slots | Description |
|---|---|
| Theme | Topic of user's command |
| aTheme | Description of topic |
| tTime | Related time of topic |
| tSpace | Related space of topic |
| object | Topic objects |
| oIdentity | Object description |
| oCardinality | Number of objects |
| oTime | Related time of object |
| oSpace | Related space of object |

**Table 3.** Examples of query commands with CURRL.

| QUERY COMMAND | CURRL |
|---|---|
| What is St. John's University (SJU), Taiwan? | Query [Theme = +SJU, aTheme = Taiwan, tSpace = At (WWW)] |
| Anything else? | Command [ Theme = +Anymore, tTime = Now, object = Related, oSpace = At (Last-one) |
| Retrieving Taiwan SJU webpages with the exception of New York SJU | ConditionalCommand [ Condition [Theme = −N.Y. SJU, tTime = Now, tSpace = At (WWW)], Command [Theme = +Taiwan SJU, tTime = Now, tSPace = At (WWW)]] |

*3.5. Establishment of System Front-End APP*

This system APP used EzoAPP (https://ezoui.com/app/) as its main development tool. However, with wearable devices (such as SNOY SWR 50 in Android Wear), Android Studio is used for development, including using a Sony SWR 50 acceleration sensor to calculate the number of steps, a graphical display of users' basic data to calculate the total calories burned on the day, and other corresponding information services. When the user executes this system APP, the system will automatically compare the current GPS position with the back-end server address to provide the administrative area information corresponding to the user's address and start the system APP operation process: (1) When the user selects the corresponding travel event, as shown in Figure 1c, the system APP converts the obtained longitude and latitude data into a corresponding complete physical address through the geocoder API via reverse geocoding [6]. (2) The system APP cuts out the keywords of the administrative area of the three-tier address. (3) The system APP sends the keywords with the CURRL in JSON format to the cloud server CEOntoIAS, and then the UAI-based LOD access technology is used to return the corresponding LOD data with the support of the aforementioned three-stage intelligent decision-making. (4) The system APP interface displays the corresponding information about the open links in the administrative areas (e.g., information about restaurants and parking lots related to "food" in general events, information about sports centers or places in the "entertainment" section, information about police and fire control in emergencies, information about medical institutions in disease or accidents). (5) When the user selects the interested information option, the detailed information (e.g., restaurant official website) will be displayed. If the user likes its content, it can be directly shared with Facebook and feedback recommendation information to enhance the robustness of LBS information in the proposed system. Currently, the front end of the proposed system includes three subagents: personal health information monitoring and consulting

agent, restaurant information consultation and sharing agent, and beverage safety consultation agent. Figure 5 illustrates their individual system architectures and functions.

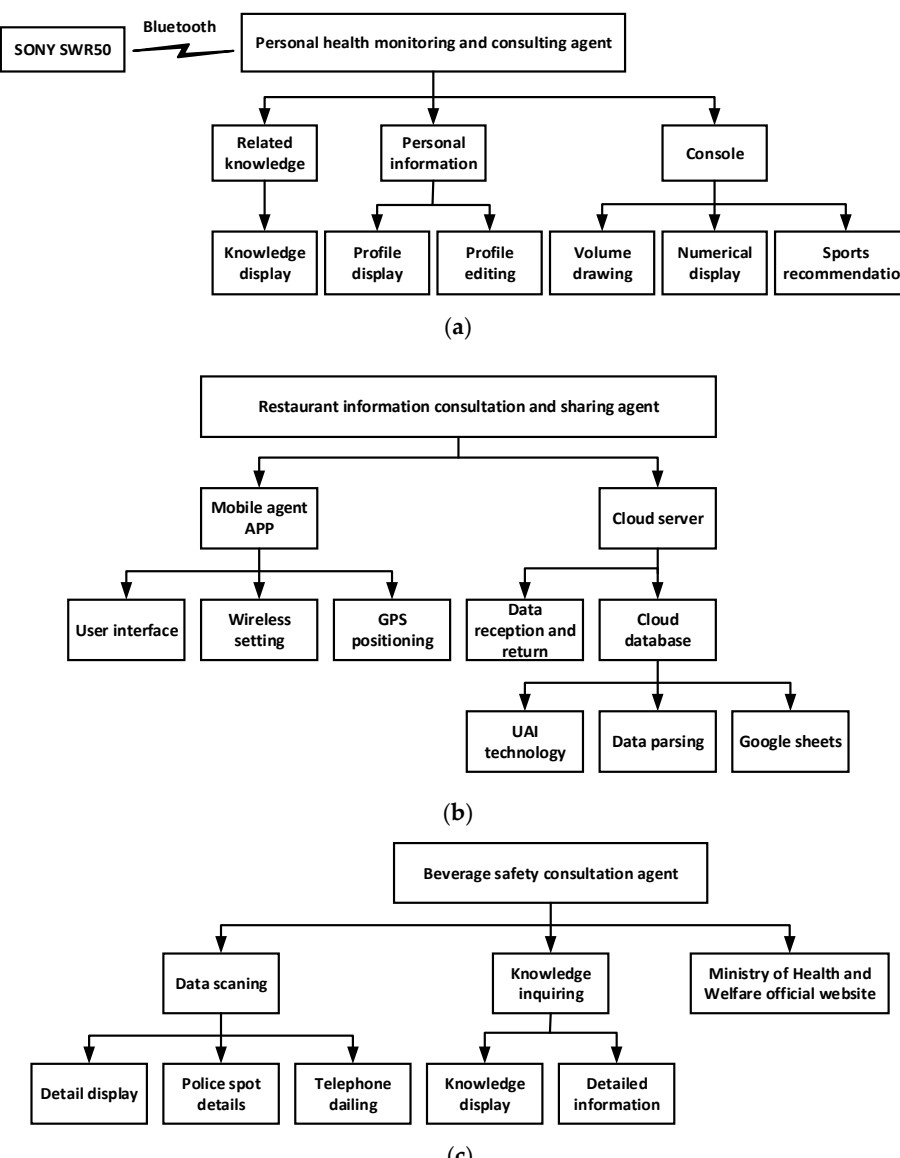

(a)

(b)

(c)

**Figure 5.** Three subsystem architectures and their functions: (**a**) personal health information monitoring and consulting agent; (**b**) restaurant information consultation and sharing agent; and (**c**) beverage safety consultation agent. GPS: global positioning system, UAI: universal application interface, APP: Application.

## 4. Evaluation of System Presentation and Efficacy

### 4.1. System Presentation

As mentioned above, the proposed system includes three basic and important tourism-related agents: the PHI agent, the RI agent, and the BS agent. On the basis of the development and practice of the Dr. What-Info system, this proposed system is expected to develop from the relevant practical technologies of mobile devices with the support of the powerful information agent system CEOntoIAS, at the back end, by referring to the UAI-based LOD access technology, as mentioned above in collaboration with GPS location interception, and with the support of appropriate link open information and domain ontology, in order to compare and intercept appropriate location service

information to effectively improve the quality of the system's mobile information regarding monitoring and recommendation.

The PHI agent [34] is used by Sony SWR50 to transfer the users' walking steps back to the system front-end APP, and then graphically display the information through the APP. The current system can display the number of steps taken in the last seven days, the average number of steps per day, the amount of calories burned today, and the number of steps taken. When the user's activity on the day is lower than the target value, the user can click the "Recommendation" button, and then, click "Related Knowledge" and "Personal Data" to switch to the selected page for the corresponding information service, as shown in Figure 6.

The information about quality restaurants and their corresponding parking lots, as recommended by the RI agent [35], is based on St. John's University GPS, as shown in Figure 7a,b, respectively. The corresponding webpage of the click on each line of the lower part of the above webpage is shown in Figure 7c, through which users can obtain all relevant information in advance. Finally, if users choose a destination, the system will automatically navigate and present the destination, as shown in Figure 7d. All the above bulk information recommendations are the greatest contributions of the extended exploration of this study.

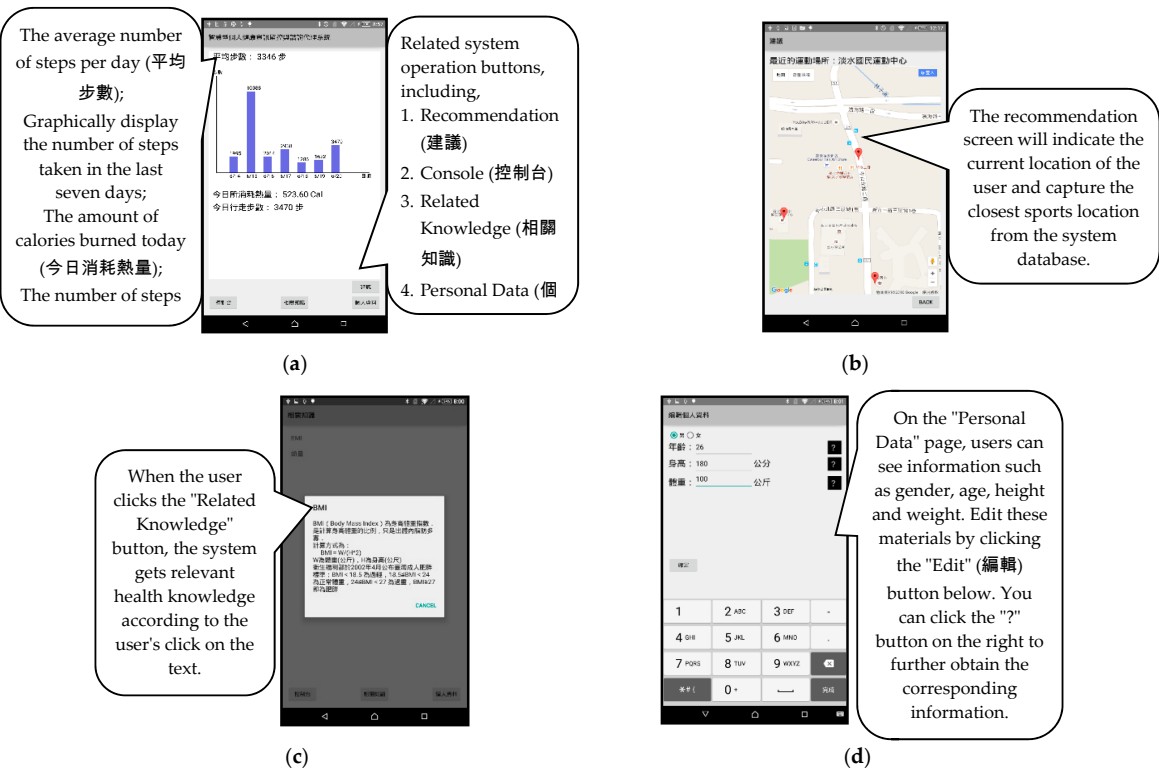

**Figure 6.** System screen display of the personal health information (PHI) agent: (**a**) system starting screen; (**b**) display of the recommendation screen at the corresponding Google map; (**c**) display of the related knowledge—BMI: Body Mass Index; and (**d**) display of the personal data—editing page.

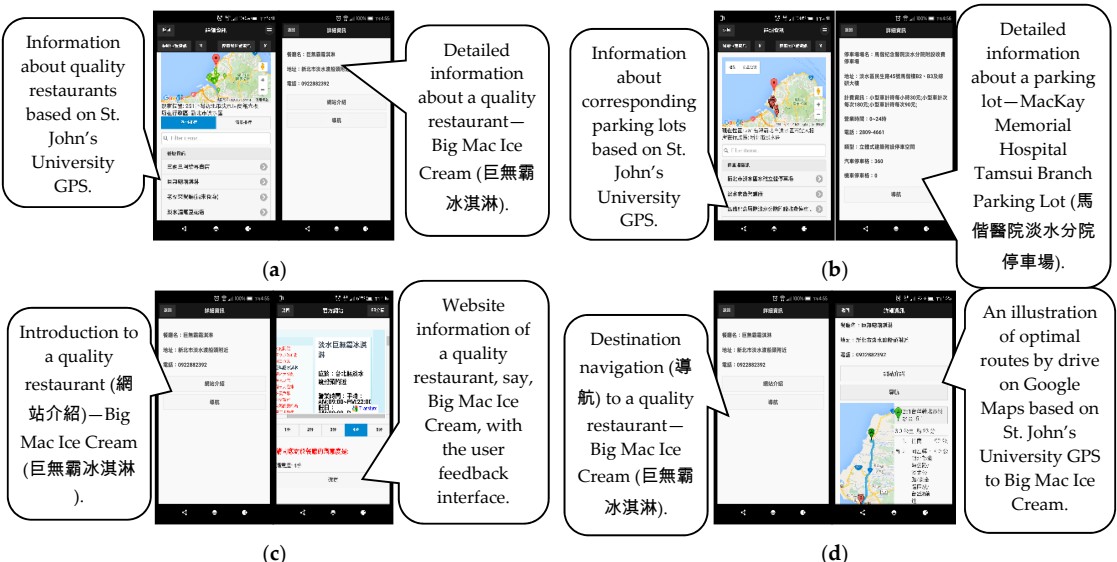

**Figure 7.** Bulk recommendation by the restaurant information (RI) agent based on St. John's University GPS: (**a**) information about quality restaurants; (**b**) information about the corresponding parking lot; (**c**) information on the website of Big Mac Ice Cream; and (**d**) destination navigation and presentation.

As mentioned above, the BS Agent [36] has two kinds of query modes—automatic scanning and manual query, as shown in Figure 8. When the user clicks the "I want to scan" button to enter the scanning page, and then presses the "Scan" button to focus on the desired QR/barcode, the system will display the decoded result on the screen and provide corresponding and appropriate information based on the system database. If the user clicks the query button of the additive, a more detailed information query action can be performed; if the user has further need, they can press the "Help needed" button to seek appropriate emergency assistance. When the user enters the emergency assistance function, the system provides a map of local police and fire stations in the case of New Taipei City, which includes current positioning information. As a result, the users can seek nearby help, and directly dial the phone to make an emergency rescue. In addition, when the user clicks the button of "Manual Query" on the homepage of the system, the user can manually activate the query keyword function. Finally, the official website of the Ministry of Health and Welfare is also provided on the homepage of the system, which is convenient for users to query.

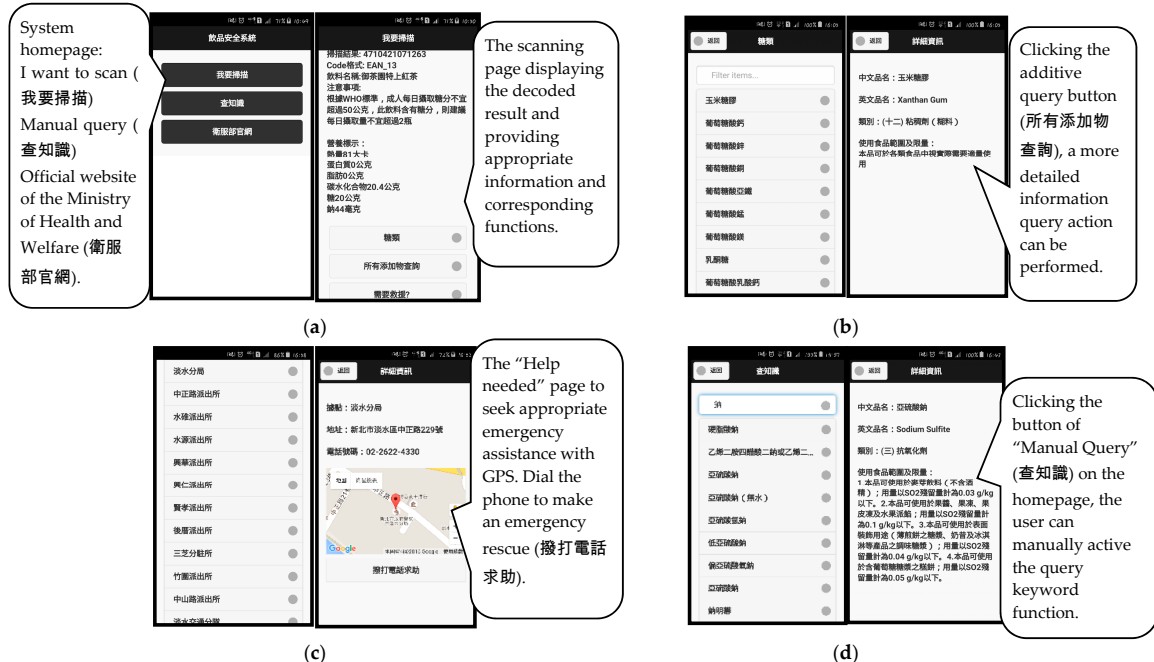

**Figure 8.** System screen display of the beverage safety (BS) agent: (**a**) system homepage and scanning page; (**b**) the additive query page; (**c**) "Help needed" page; and (**d**) "Manual Query" page.

## 4.2. System Evaluation

The following comparisons and experiments invited users from related departments (department of information and communication), unrelated departments (department of business management), and the middle/high age-bracket (users above 45 years of age). There were 10 persons in each group, and the 3 groups were used as a reference frame for analysis. All comparisons and evaluation results were based on opinions in which there was a 75% or greater agreement in the three groups, which represented that the majority of the people were in agreement.

The three main comparison objects of the PHI agent are similar to personal mobile APPs, such as Google Fit, Millet Movement, and Motion-Pedometer [34]. Table 4 shows the comparison results of the "basic display" functions of the related system. The corresponding comparison systems are not much different from the proposed system, and the parts of the function display are similar. Table 5 illustrates the "advanced query" functions between the proposed system and the control system, showing the main wisdom of the proposed system that can recommend suitable health information in a timely and appropriate manner. In summary, the three control APPs are displayed in a bar graph when displaying the number of steps in its most recent use. The Millet Movement must be bound to the Millet bracelet to be applied, meaning its system cannot be paired with other wearable devices. The motion-pedometer requires a fee to activate other advanced functions, such as recommendation functions. However, the aforementioned functions, namely the mobile APP proposed by the PHI agent, can be easily solved, which is completely in line with the original intention of this research—timely, appropriate, and free, and the latest and complete mobile information services.

**Table 4.** Comparison results of system basic functions.

| | Google Fit | Millet Movement | Motion-Pedometer | PHI Agent |
|---|---|---|---|---|
| User profile | ✓ | ✓ | ✓ | ✓ |
| Display method | Line chart/bar graph  | Bar graph  | Bar graph  | Bar graph  |
| Query the number of steps in the day | ✓ | ✓ | ✓ | ✓ |
| Query the calories on the day | ✓ | ✓ | ✓ | ✓ |

Legend: "✓" means it has this function, whereas "**x**" means it does not.

**Table 5.** Comparison results of system advanced functions.

| | Google Fit | Millet Movement | Motion-Pedometer | PHI Agent |
|---|---|---|---|---|
| Query the number of steps in the month | ✓ | ✓ | **x** | **x** |
| Query the calories in the month | ✓ | ✓ | **x** | **x** |
| Related knowledge | **x** | **x** | **x** | ✓ |
| User group | **x** | **x** | ✓ | **x** |
| User profile | **x** | **x** | **x** | ✓ |
| Recommendation | No | No | Paid to activate | Inform the user of the closest sports venue on the basis of location of the user |

Legend: "✓" means it has this function; whereas "**x**" means it does not.

The RI agent effectiveness experiment first researched the system's effectiveness and various function tests. This study used the comparison table for LBSs, as proposed by Tu [37], and classified the proposed 12 items into location positioning, hardware device, application management, and system presentation for comparison [35]. This experiment indicated that a sound information recommender system must conform to more verification conditions; therefore, the information recommendation and value added service classes were added to the comparison table, as shown in Table 6. The object system compared in this experiment was the LBS system developed by the government—Taiwan Food Map (http://www.foodmap.tw/FM.html). As the system quality matches this system, it was selected as the control group. The overall function's up-to-standard rate of this system was 87.5%, whereas the up-to-standard rate of the Taiwan Food Map, as developed by the government, was only 43.7%. The reasons were that (1) this system uses location information efficiently to provide a more accurate

information service, and (2) with the support of LOD (linked open data), this system architecture uses the back-end database more efficiently and provides users with more all-inclusive LBS.

**Table 6.** System function comparison analysis.

| Class | Item | RI Agent | Taiwan Food Map |
|---|---|---|---|
| Location positioning | Indoor positioning | ✓ | x |
| | Outdoor positioning | ✓ | x |
| Hardware device | RFID storage | x | x |
| | Environmental sensing device | x | x |
| Application management | DBMS | ✓ | ✓ |
| | Management application | ✓ | ✓ |
| | Mobile app | ✓ | x |
| | Data transmission | ✓ | ✓ |
| System presentation | Easy update | ✓ | ✓ |
| | Ease of implementation | ✓ | ✓ |
| | Actual display | ✓ | ✓ |
| Information recommendation | Location based service (LBS) | ✓ | ✓ |
| | Community sharing | ✓ | x |
| | Satisfaction | ✓ | x |
| Value added service | Emergency call | ✓ | x |
| | Navigation | ✓ | x |

Legend: "✓" means that it has this function; whereas "x" means it does not.

Regarding the comparison of the BS agent functions [36], the authors selected the iQC Commodity Safety Information website app (http://iqc.com.tw/). This app provides free use of commodity barcodes for the public to check whether a product has passed inspection, and provides qualified information. The iQC can also manually specify the type of product to be queried through keywords. The important thing is that the source information of the product is the open data of the open data platform of the Health and Welfare Department, Taiwan, which is also used as a source of data through cloud link technology, and regularly updates its data content regarding products. Summarizing the above similarities, iQC is used as a comparison system to verify the pros and cons of the proposed system. Furthermore, it is necessary to evaluate the overall usability of the two-system interface design, especially the usability of the interface content and the ease of use of system operation. The analysis of the former is based on the International Organization for Standardization (ISO) 9241; according to its definition, three dimensions—effectiveness, efficiency, and satisfaction are used to evaluate usability. Whitney Quesenbery (http://www.wqusability.com/) suggested using the "5E" function, which measures usability. Regarding the ease of using the system, Jakob Nielsen (http://www.useit.com/jakob/) strongly recommended 10 basic principles for user interface design. The relevant evaluation parameters and the analysis results are shown in Table 7. There are 15 items in the overall interface satisfaction, which are calculated as 7 points for each item, where full satisfaction is 100 points, over 100 points for calculating as 100 points. The satisfaction score of the proposed system interface design is 84 points, whereas the iQC App score is 56 points. The main differences between the two systems are (1) the system takes advantage of considering the humanized guidance design concept, which directly enhances the visibility of the system in terms of affinity, accessibility, transparency of system status, and avoidance of errors, and allows users to identify the extent of their work without back-tracking, and (2) both systems have room for improvement in two aspects: error tolerance and help documents.

**Table 7.** Analysis of overall satisfaction with interface design of the two systems.

| | Standard | Items | BS Agent | iQC App |
|---|---|---|---|---|
| Usability | Quesenbery5E | Efficient | ✓ | ✓ |
| | | Effective | ✓ | ✓ |
| | | Engaging | ✓ | x |
| | | Error tolerant | x | x |
| | | Easy to learn | ✓ | ✓ |
| Easy to use | Nielsen | Visibility of system status | ✓ | x |
| | | Match between system and the real world | ✓ | ✓ |
| | | User control and freedom | ✓ | ✓ |
| | | Consistency and standards | ✓ | ✓ |
| | | Error prevention | ✓ | x |
| | | Recognition rather than recall | ✓ | x |
| | | Flexibility and efficiency of use | ✓ | ✓ |
| | | Aesthetic and minimalist design | ✓ | ✓ |
| | | Help users recognize, diagnose, and recover from errors | x | x |
| | | Help and documentation | x | x |
| | | Total satisfaction score | 84 | 56 |

Legend: "✓" means it has this function; whereas "x" means it does not.

The design preference to importance ratio (DIR) (Equation (1)), as proposed by Ha [38], was taken as the design principle of the human–machine interface (HMI) of this system, as it is perfect for combining with the balancing index (BI) (see Equation (2)) to define the interface. In other words, if BI is zero, all the HMI elements in this design are balanced and perfect. Its physical meaning is that the HMI design meets the principle of the design preference for importance, and that users' operation of the system is more consistent with user demand:

$$DIR_{ijk} = \frac{\frac{DP_{ij}}{\sum_{i=1}^{n} DP_{ij}}}{\frac{I_{ik}}{\sum_{i=1}^{n} I_{ik}}}, \tag{1}$$

$$BI_{jk} = \frac{\left| \sum_{i=1}^{n} \log_{10} DIR_{ijk} \right|}{n}, \tag{2}$$

where $DIR_{ijk}$ refers to the DIR of design attribute $j$ and importance attribute $k$ of HMI interface element $i$. $DP_{ij}$ indicates the design preference of design attribute $j$ of HMI interface element $i$. $I_{ik}$ represents design importance $k$ of the importance attribute of the HMI interface element $i$. $BI_{jk}$ refers to the balance index of design attribute $j$ and importance attribute $k$, whereas $n$ indicates the total number of HMI interface elements.

A five-scale evaluation scheme was utilized to evaluate each design preference ($DP_{ij}$). In other words, the design preferences included very good, good, moderate, weak, and very weak, and their corresponding values were 5, 4, 3, 2, and 1, respectively. Table 8 illustrates the evaluation of the HMI elements in our design attributes, as well as their corresponding informational importance of proposed agents, as evaluated using the analytic hierarchy process [38], whereas Table 9 shows the evaluation results of *DIR* with average *BI* = 0.008430 (*BI* should approach zero for the best balance) of the proposed agents. "A little bit needs to improve" means that the proposed system interface

design must be slightly adjusted but will not affect the operations of the whole system. The verification results showed that the human–machine interface of our proposed agents can meet important design preferences and provide approximately optimal balance.

**Table 8.** Evaluation of human–machine interface (HMI) elements and their corresponding importance of proposed agents.

| Agent | HMI Elements | Description | Design Preference | Informational Importance | Remarks |
|---|---|---|---|---|---|
| PHI Agent | RDBOT | Recommendation Bottom | 5 | 0.192308 | Bottom in text |
| | CBOT | Console Bottom | 3 | 0.269231 | Bottom in text |
| | RKBOT | Related Knowledge Bottom | 3 | 0.269231 | Bottom in text |
| | PDBOT | Personal Data Bottom | 3 | 0.269231 | Bottom in text |
| RI Agent | GMAP | Google Map | 5 | 0.166667 | Graphical display |
| | GPS | GPS Location | 3 | 0.233333 | Label in text |
| | RDBOT | Recommendation Bottoms | 3 | 0.233333 | Bottom in text |
| | NAVI | Navigation | 3 | 0.183333 | Bottom in text |
| | FB | Sharing by Facebook | 3 | 0.183333 | Label in text |
| BS Agent | SBOT | Scanning Bottom | 5 | 0.263158 | Bottom in text |
| | QBOT | Query Bottom | 3 | 0.368421 | Bottom in text |
| | WBOT | Website Bottom | 3 | 0.368421 | Bottom in text |

**Table 9.** Evaluation results of design preference to importance ratio (*DIR*) with average balancing index (*BI*) = 0.008430 of proposed agents.

| Agent | HMI Elements | DIR | Description | BI | Average BI |
|---|---|---|---|---|---|
| PHI Agent | RDBOT | 1.303242 | A little bit needs to improve | 0.016884 | 0.008430 |
| | CBOT | 0.926256 | A little bit needs to improve | | |
| | RKBOT | 0.930887 | A little bit needs to improve | | |
| | PDBOT | 0.926256 | A little bit needs to improve | | |
| RI Agent | GMAP | 1.266055 | A little bit needs to improve | 0.003165 | |
| | GSP | 0.904325 | A little bit needs to improve | | |
| | RDBOT | 0.904325 | A little bit needs to improve | | |
| | NAVI | 1.000834 | Approximately optimal balance | | |
| | FB | 1.000834 | Approximately optimal balance | | |
| BS Agent | SBOT | 1.270882 | A little bit needs to improve | 0.005242 | |
| | QBOT | 0.903256 | A little bit needs to improve | | |
| | WBOT | 0.903256 | A little bit needs to improve | | |

Finally, the system effectiveness experiment aimed to analyze the actual operation and experience satisfaction of the proposed system APPs. The three aforementioned groups were used as a reference frame for analysis, as shown in Table 10. The satisfaction scale was 1–5 points, representing strongly dissatisfied (1 point) to most satisfied (5 points). It was observed that unrelated departments care more about the interface appearance and the operation of this system being more convenient. The effect was significant in the middle/high age-bracket. Related departments indicated that operation and fluency are somewhat insufficient and should be enhanced. An important indicator was that information correctness was graded as being high. According to this table, fluency is somewhat insufficient, but the graphical menu design of the system APP is easy for the users of the middle/high age-bracket to operate, which is an obvious outcome.

**Table 10.** System interface design and overall satisfaction analysis.

| Topic | Item | Related Departments | Unrelated Departments | Middle/High Age-Bracket |
|---|---|---|---|---|
| Interface | Interface design | 3.5 | 2.5 | 3.8 |
| | Whether the required function is found | 3.9 | 3.7 | 3.6 |
| Operation | Operation difficulty | 4.2 | 3.9 | 4.2 |
| | Operation fluency | 3.3 | 3.1 | 3.8 |
| Function | If function is perfect | 3.2 | 3.9 | 4.0 |
| | If function is practical | 4.1 | 3.8 | 3.1 |
| Information | If information is correct | 4.5 | 4.3 | 4.5 |
| | If information recommendation is perfect | 3.6 | 3.9 | 3.7 |

## 5. Conclusions and Discussion

On the basis of wearable devices, web services, and linked open data, this study's intelligent travel cloud information monitor and recommendation multi-agent system explored LOD access technologies for OpenData@Taiwan by integrating the aforementioned UAI technology links for increasing the accuracy, authenticity, and integrity of intelligent tourism information, effectively adding value to the quality of cloud information monitoring and recommendation through the support of three-stage intelligent decision-making with CEOonoIAS. This paper extends the application scope of Dr. What-Info and reached the ultimate goal of the research—the construction of an intelligent mobile information monitoring and recommendation multi-agent system with friendly interfaces for autonomously providing corresponding cloud information. The results and conclusions of the system's practical exploration are listed as follows:

(1) This study established three-tier address-based, UAI-based LOD access technology by actively developing the proposed system with friendly interfaces that assist users in quickly, accurately, and effectively obtaining relevant and timely useful links and open information.

(2) With the operational support of CEOntoIAS, this study established a cloud interactive paradigm supported by web services through the corresponding query semantic content with the CURRL in JSON format. With the support of three-stage intelligent decision-making, such as OntoDMA, OntoCBRA, and OntoIAS, the proposed system APPs autonomously and effectively added value to the quality of the monitoring and to the recommendation of intelligent tourism information. Therefore, this study is unique and different from other information agents found in the literature.

Furthermore, regarding the evaluation results mentioned above, the mobile APP, as proposed in this system, can be easily solved, which is completely in line with the original intention of the research—timely, appropriate, and free, as well as providing the latest and complete mobile information services. In addition, based on Tu's comparison table for LBSs, the proposed system receives an overall function up-to-standard rate of 87.5%, and such recommendations provide users with high information correctness and user satisfaction. The proposed system architecture not only uses the back-end database more efficiently; it also provides users with more all-inclusive LBS. The satisfaction score of the usability of the proposed system and the ease of using the interface design was 84 points in terms of Quesenbery's 5Es and Nielsen ratings, which indicated that this system APP must be improved in two aspects: error tolerance and help documents. The verification results of the interface design showed that the human–machine interface of our proposed system can meet important design preferences and provide approximate optimal balance with average $BI = 0.008430$. Finally, although the interface appearance is more convenient, the operation and fluency are somewhat insufficient, and should be enhanced. Information correctness was graded as being high, and the graphical menu design of the system APP was easy for users to use, which were obvious outcomes. Although there

is plenty of room for improvement in experience and in more travel-related agents, such as instant traffic assistance agents and trip planning agents, the feasibility of the proposed service architecture has been proven.

**Author Contributions:** This research article contains two authors, including K.-Y.C. and S.-Y.Y.; K.-Y.C. and S.-Y.Y. jointly designed the overall architecture and related algorithms and also conceived and designed the experiments; however, Prof. S.-Y.Y. coordinated the overall plan and direction of the experiments and related skills. K.-Y.C. and S.-Y.Y. not only contributed the analysis tools but also analyzed the data. K.-Y.C. performed the experiments and S.-Y.Y. wrote this paper and related replies.

**Acknowledgments:** The authors would like to thank Yu-Jui Tsai, Yong-Han Tsai, and Chi-Ling Ku for their assistance in the earlier system implementation and preliminary experiments. This research was sponsored under grant 106-2221-E-129-008 and 107-2632-E-129-001 by the Ministry of Science and Technology, Taiwan, R.O.C. The authors feel deeply indebted to the Department of Information Management and the Department of Information and Communication at St. John's University, Taiwan, for all aspects of assistance provided.

**Conflicts of Interest:** The authors declare no conflict of interest.

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
