# Peer review of "A Cloud Information Monitoring and Recommendation Multi-Agent System with Friendly Interfaces for Tourism"

_applsci, doi:10.3390/app9204385_

Round 1
Reviewer 1 Report
This study focuses on presenting a methodology to discover the current technology trends and user information requirements for tourism. A System Technology, Architecture and Operation is proposed to handle it.
Overall evaluation:
The paper is very confusing. The proposal has not been well defined and specified. The contributions are unclear, and even its purpose is vague. The decisions made need further explanation and justifications, and there are some unclear parts and aspects that must be explained and better described.
Comments:
1. The title of this paper is very confusing. I suggest a shortened and more concise title. Indeed, the whole paper is confusing. There are no-sense paragraphs and sentences (i.e. lines 61-62). There are too long sentences which difficutl reading the document and undertanding it (i.e. lines 217-224).
2. Abstract must be rewritten. The abstract is there to grip the reader. I suggest that the abstract is written in the following format (in the same paragraph):
[Background] How does the research relate to applies sciences
[Reason for study] What was the rationale/aims of the study
[Research Design] Mention the research design and paradigm
[Data collection] Data gathering tools
[Findings] Short Summary of the main findings
[Conclusions] Briefly indicate the implications of the findings
3. The web references into the text should be avoided.
4. I don’t think that “MBA Think Tank” is a pretigious actor in tourism. Instead, you should use bibliographic cites to support your assumptions.
5. The terms “smart tourism” and “smart earth?” should be defined.
6. I think that the predictions by advisory companies and other personal opinions can not be placed in a scientific work. In general, they do not provide rigurous data (obtained trough scientifc methodology).
7. Acronyms must be defined before use them in the text.
8. The aims of the work are unclear. The contributions to the state of the art are not clearly described.
9. Most figures are illegible. (i.e. fig 1, 2, 3, …)
10. The proposal should be structured in order to explain the proposed system and the aspects needed to build it. For example, there are a lack of understanding for several basic aspects of the proposal: what is the need of an ontology?, what is the role of this tool in your system? What is “UAI-based LOD cloud database”? What is the purpose of it?
11. What is the relation between the WIAS concept and your system?
12. Why has the system only three subsystems? Why these ones?
13. The comparison with other Apps should be made with similar applications. In this sense, is your system a “physical activity monitoring” App? If answer is no, then the comparison made in tables 3 and 4 does not make any sense.
14. How have been obtained the data of tables 7 and 8? Have you made a survey to users of your system? How have you made the evaluation?
…
Reviewer 2 Report
The authors propose a travel cloud information consultation and sharing multi-agent system.
The proposed approach is really interesting but there are some points to be clarified.
Some experiments on efficiency of the approach should be added. Furthermore, the authors should provide more details about the used technologies.
The related work should be extended with other recommendation approaches. See for examples:
1) Recommendation in social media networks. In 2017 IEEE Third International Conference on Multimedia Big Data (BigMM) (pp. 213-216). IEEE.
2)Kira: a system for knowledge-based access to multimedia art collections. In 2017 IEEE 11th International Conference on Semantic Computing (ICSC) (pp. 338-343). IEEE.
Finally a linguistic revision is necessary.
Reviewer 3 Report
In the introduction is not specified what are the contributions of the authors presented in the article.
In Introduction, must contains discussions about the problem could be improved with the examples.
The work methodology must be presented much clearer in order to be accepted for publication.
Round 2
Reviewer 1 Report
The paper has been improved and most of my concerns have been addressed. However, some importat issues still remain. In general there is a lack of rigor. Sometimes you write more text than necessary, but others it would be necessary to better explain and justify the decisions made.
For example1, you said in line 50:
“Gartner announced the top 10 technology trends [...]”
Who is Gartner?, What is its credibility? What is its relevance in tourism sector?
For example2, you said in line 53:
“according to IBM, big data has four characteristics (the 4V of big data) […]”
Why do you not use scientific references for this definition?
(In this reference you have numerous definitions of BigData: https://pdfs.semanticscholar.org/bdd8/0f0da1b5ac439cf90e800a667e34094a130c.pdf)
For example3, you said in line 84:
“The rise of the IoT (or Internet of Everything, IoE)…”. IoT and IoE are not the same.
For example 4, you said in line 85:
“According to Gartner5…” and 5 is this link: http://iknow.stpi.narl.org.tw/Post/Read.aspx?PostID=15059 . What kind of information is this?. It is not in English and is not valid to be referenced here. You should use verificable references and links for a wide audience. (the same for #6 link).
You use “Protégé” tool. why?
These are only some examples of lack of rigor.
In addition, some text remain unclear. For example: “The most important driving force for this research to cite the above four driving forces explore a cloud information monitoring and recommendation multi-agent system”. That sentence has no sense. Whay is that importat driving force?
For example2, what does “Dr. What-Info” mean?
The overall contributions of this paper is not entirely clear. You state in the abstract:
“This paper further develops a correct, authentic, and integrated multi-agent system for tourism cloud information monitoring and recommendation based on wearable devices, GPS, and linked open data technology, representing the establishment of an intelligent mobile information monitoring and recommendation multi-agent system with the characteristics of precision, rapidity, toughness, universality, and initiative”.
It is a very long definition of your work. Some words are repeated. What does “authentic” mean? What does “toughness” mean?, and “universality”?, and “initiative”?.
I recommend to use a short description of your work with the key aspects that define its main contributions.
For example (this is just a suggestion):
“This paper develops a novel citizen-centric multi-agent information system for tourism sector”.
-The term “novel” emphasizes the novelty of the proposal.
-The term “citizen-centric” suggests that wearables or mobile devices are used and IoT paradigm is included.
- The term “multi-agent information system” shows the core technology used in the construction of the system. It suggest that a ontology is needed.
-The term “tourism sector” shows applicability and provides credibility to the proposal.
The remain features such as correct, precise, tough, … are supposed to be included in the proposal by default. It is not needed to mention them in the description of the contribution. They will be explained in the paper.
Reviewer 3 Report
In my opinion, the paper can be accepted in this form. Some minor revision of the presentation mode can be performed for a greater clarity.
Round 3
Reviewer 1 Report
The authors have addressed all my main concerns. This version of the document greatly improves the first submission.
Some minor comments:
-Figure 2 can be dropped. I think It does not provide valuable information.
